# Non-Invasive Detection of Biomolecular Abundance from Fermentative Microorganisms via Raman Spectra Combined with Target Extraction and Multimodel Fitting

**DOI:** 10.3390/molecules29010157

**Published:** 2023-12-27

**Authors:** Xinli Li, Suyi Li, Qingyi Wu

**Affiliations:** 1College of Instrumentation and Electrical Engineering, Jilin University, Changchun 130061, China; 2Changchun Institute of Optics, Fine Mechanics and Physics, Changchun 130033, China; 3University of Chinese Academy of Sciences, Beijing 100049, China

**Keywords:** biomolecular abundance detection, fermentative microorganisms, Raman spectra, target extraction, multimodel fitting

## Abstract

Biomolecular abundance detection of fermentation microorganisms is significant for the accurate regulation of fermentation, which is conducive to reducing fermentation costs and improving the yield of target products. However, the development of an accurate analytical method for the detection of biomolecular abundance still faces important challenges. Herein, we present a non-invasive biomolecular abundance detection method based on Raman spectra combined with target extraction and multimodel fitting. The high gain of the eXtreme Gradient Boosting (XGBoost) algorithm was used to extract the characteristic Raman peaks of metabolically active proteins and nucleic acids within *E. coli* and yeast. The test accuracy for different culture times and cell cycles of *E. coli* was 94.4% and 98.2%, respectively. Simultaneously, the Gaussian multi-peak fitting algorithm was exploited to calculate peak intensity from mixed peaks, which can improve the accuracy of biomolecular abundance calculations. The accuracy of Gaussian multi-peak fitting was above 0.9, and the results of the analysis of variance (ANOVA) measurements for the lag phase, log phase, and stationary phase of *E. coli* growth demonstrated highly significant levels, indicating that the intracellular biomolecular abundance detection was consistent with the classical cell growth law. These results suggest the great potential of the combination of microbial intracellular abundance, Raman spectra analysis, target extraction, and multimodel fitting as a method for microbial fermentation engineering.

## 1. Introduction

The physiological status of single cells is a major factor in determining the yield of fermentation products in fermentation engineering [1]. With the increasing cost of fermentation raw materials, and increasing environmental awareness, the precise regulation of the fermentation process is highly demanded to improve the quality of products. The abundance of microbial molecules (e.g., proteins, nucleic acids, etc.) is closely related to the regulated fermentation process [2]. Therefore, the accurate detection of the abundance of the biomolecules within fermentative microorganisms is significant in the fermentation process.

Traditional biomolecular detection techniques mainly include fluorescence [3], mass spectrometry [4], polymerase chain reaction (PCR) [5], and so on. However, these techniques cannot provide the heterogeneity of single cells and their rich fingerprint information of biomolecules. Hence, the development of a method for biomolecular abundance detection at the single-cell level is urgently needed to fully reveal the heterogeneity during the microbial fermentation process.

The Raman spectrum is a fingerprint spectrum that provides relatively high sensitivity and is widely applied in the biological field [6,7]. Notably, this non-invasive strategy can be used for the detection and analysis of single-cell stress responses, which can reveal the structure, conformation, and other abundance information of biomolecules in living cells [8]. Therefore, this non-invasive technique provides a new possibility, to study the abundance of biomolecules within fermentative microorganisms. However, the Raman peaks of biomolecules are complex, and relatively weak, with spectral peak superposition distortion, peak position micro-shift, and small peak flooding, resulting in inaccurate analysis. Consequently, Raman spectra require advanced data processing to extract meaningful information, due to their complexity and heterogeneity.

Machine learning provides an unprecedented opportunity to extract information from complex or large datasets. Importantly, machine learning shows superior performance in analyzing Raman spectra signals from complex biological samples [9,10], and can be trained to recognize features in Raman spectra and assign them to the proper label, corresponding to the identity of the analyte. The XGBoost model, as a machine learning algorithm, possesses excellent interpretability and outstanding feature recognition ability of the tree structure [11]. Consequently, it can be useful to identify the potential relationship between intracellular biomolecules and their characteristic peaks during bacterial growth.

Here, a non-invasive detection method was developed to study the biomolecular metabolic changes of fermenting microorganisms at the single-cell level during different growth stages. The characteristic Raman peaks associated with cell growth markers were extracted by calculating the high gain information of each characteristic of optimized XGBoost at the cell growth division nodes. Meanwhile, the single-peak areas were calculated using total peak area (TPA), and a Gaussian multi-peak fitting algorithm was developed to calculate split-peak areas to improve the accuracy of biomolecular abundance calculations. The recognition accuracy was 94.4% and 98.2% during different culture times, and cell cycles of bacterial growth, through the XGBoost algorithm. The ANOVA analysis results can realize more accurate and intuitive abundance detection of proteins, nucleic acids, and other biomolecules during single-cell growth. Raman spectra, combined with target extraction and multimodel fitting, provide an accurate and reliable method for the detection of the abundance of biomolecules within fermentative microorganisms, which has an important guiding role in the precise regulation of the microbial fermentation process.

## 2. Results and Discussion

### 2.1. Characterization of the Bacterial Growth Process

To assess the growth cycle of *E. coli*, the OD values at 600 nm were tested using the UV–Vis spectra during different culture times (Figure 1a). From Figure 1a, the OD values of the bacterial solution were gradually increased, and the incubation time was prolonged. Meanwhile, the lag phase of *E. coli* was observed after incubation for 2 h based on the OD value at 600 nm. Confocal Raman spectral microscopy was performed, to observe the brightfield image of *E. coli*, as shown in Figure 1b. Figure 1c depicts the bright field image of *E. coli* in the lag phase. It indicates that the speed of cell division was relatively slow, resulting in lower proliferation (Figure 1a,c). The *E. coli* showed rapid proliferation to present an exponential increase after incubation time from 3~6 h (Figure 1a,d), indicating that the cells were in the log phase. After an incubation time of 7 h, the OD values of the bacterial solution at 600 nm basically remained the same (Figure 1a), which indicated that cell proliferation and apoptosis were in a dynamic equilibrium stage. Therefore, we considered the cells in a stationary phase after the incubation time of 7 h, and the corresponding bright-field image of *E. coli* is shown in Figure 1e. A detailed description of bacterial culture and Raman detection is given in Appendix A. In this work, we have checked the biomolecule abundance detection at the lag, log, and stationary phases, respectively. Subsequently, the Raman spectrum was applied to discriminate the abundance of the biomolecules, which can provide rich fingerprint information [12,13]. To achieve an accurate measurement of biomolecule abundance, the Raman mapping was performed to detect the distribution of the Raman intensity at 1576 cm^−1^, as shown in Figure 1f,g. From Figure 1g, the Raman mapping indicated that the distribution of Raman intensity at 1576 cm^−1^ showed heterogenicity in the whole cell. To reduce the effect of intracellular spatial heterogeneity on the detection of biomolecular abundance, the Raman signal was measured at three equally spaced points along the length of a bacterium, and their average spectra were calculated in the next study.

### 2.2. Detection of the Growth Process of E. coli Using Raman Spectra

To detect the abundance of the biomolecules within *E. coli* during the growth process, the Raman spectra of *E. coli* at different culture times (0, 1, 2, 3, 4, 5, 6, 7, 8, 16, and 24 h) were collected, as shown in Figure 2a. Subsequently, to assess the stability of the Raman spectra of *E. coli* during the growth process, signal-to-noise ratio (SNR) values were calculated using the maximum peak intensity above the baseline divided by the noise intensity in the silent region around 1800 cm^−1^. From Figure 2b, the boxplot with jitter points shows the SNR distribution and the spectral SNR values are concentrated between three and seven. Further, we used the Kolmogorov–Smirnov (KS) normality of the SNR density curves (Figure 2c) to estimate the stability of the data, which showed that all 12 curves followed a normal distribution, and the *p*-value was 8.12 ± 1.63 >> 0.05. In summary, the SNR result shows a more stable distribution, indicating that the expected detection results are not affected by SNR.

### 2.3. Target Extraction of Active Biomolecules during Cell Growth Process

To observe the distribution of cells in different growth cycles, the clustering analysis of the Raman data was conducted through the principal component analysis (PCA) method and retained 95% of the characteristic contribution. Further, the first two feature spaces of the linear discriminant analysis (LDA) are used to visualize the dimensionality reduction, represented by PC_LD 1 and PC_LD 2 in Figure 3a. The silhouette coefficient value for the three cell cycle clusters estimated is 0.89, which demonstrates the high cohesiveness and low coupling between the different cell cycles. Meanwhile, to train the single-cell growth detection model, peak extraction was performed using the XGBoost algorithm. The main hyperparameters were selected using grid search and 3-fold cross-validation. The base classifier model is gbtree. The number of base classifiers and tree depth are set as 90 and 7, respectively. Figure 3b shows the confusion matrix of the 20% test data, which corresponds to 11 sets of culture times. The cell cycle of *E. coli,* including the lag, log, and stationary phases, was clearly located in blue, yellow, and green regions, respectively. The average detection accuracy of the cell growth detection model based on XGBoost is 94.4% for 11 sets of culture times, and 98.2% for three cell cycles. The sensitivity and specificity of the XGBoost model were estimated using receiver operating characteristic (ROC) curves (Figure 3c). The area under the ROC curve (AUC) value is the sum of the true positive rate (TPR) and the false positive rate (FPR), [14] and all three growth cycles are above 0.99, indicating that the XGBoost model performs well. Based on this, the characteristic peaks extracted from Raman spectra were performed using the high gain algorithm, which strongly correlated with active biomolecules within intracellular metabolism.

As shown in Figure 3d, the gain ranking of the top 20 most dominant characteristic peak positions (786, 1576, 1034, 782, 1586 cm^−1^, etc.) was extracted using the XGBoost, which was associated with cell growth [15,16]. The high gain is considered a potential indicator of the effect on cell metabolism [17,18]. Consequently, these characteristic peak positions can provide the most important information about the intracellular metabolic activity of fermenting microorganisms. From the characteristic gain ranking results, we removed the duplicate characteristic peaks at 780 and 782 cm^−1^. The characteristic peaks at 810 and 940 cm^−1^ are assigned to phosphodiester, polysaccharide, and amylose, which were also removed. Therefore, the final extracted characteristic peak positions are assigned to protein and nucleic acid, as shown in Appendix A.

### 2.4. Characteristic Peak Fitting to Improve the Accuracy of Biomolecular Abundance Calculations

The areas of characteristic peaks extracted using the XGBoost were calculated using TPA, and the average value of the spectra of the three positions to the left and right characteristic peaks was used as the background value. For mixed peaks with the presence of shoulder peaks (such as 1096 cm^−1^, etc.), we used Gaussian multi-peak fitting to decompose the mixed peaks and improve the accuracy of the net peak area calculation. A nonlinear programming function was used to optimize the initial parameters of the Gaussian multi-peak fitting to improve the fitting accuracy and convergence speed. Figure 4a and Figure 4b show the peak loci and fitting curves for 726 and 1096 cm^−1^, respectively. Figure 4c shows the results of the fitting to Figure 4a using TPA, where A is the net peak area and B is the background area. Figure 4d displays the fitting to Figure 4b using the Gaussian multi-peak fitting, where the net peak area (A, 1096.8 cm^−1^) value is 1.12 ± 0.18, the standard error of fitting is 0.09 ± 0.03, and the goodness-of-fit (R2) is 0.94 ± 0.05, and B is the shoulder peak area at 1102.2 cm^−1^ in the mixed peak.

### 2.5. Visual Analysis of Biological Abundance

The area values of characteristic peaks attributed to protein and nucleic acids were calculated based on the TPA or the Gaussian multi-peak fitting algorithm. From Figure 5a,b, the three colors from left to right represent the three cell cycles of the cell growth process, respectively. Statistical analysis of significance levels was performed using ANOVA, where * represents *p* < 0.05, ** *p* < 0.01, and *** *p* < 0.001. The calculation results show that the protein abundance gradually increased with the lag, log, and stationary phases of cell growth (Figure 5a), and the nucleic acid abundance reached its maximum value in the lag phase of cell growth (Figure 5b). The average protein and nucleic acid abundances were then plotted as heat maps in Figure 5c,d. The row-normalized heat maps indicated that the protein peak increased with protein synthesis during the cell cycle. However, the nucleic acid increase was mainly concentrated in the log phase, because the peak areas increased significantly with nucleic acid replication in this phase. The relative abundance of nucleic acid in the stationary phase decreased (Figure 5b). To further prove this phenomenon, yeast was also detected using Raman spectra, and the abundance of protein and nucleic acid was also calculated, as shown in Figure 5e,f. The trend of abundance changes of protein and nucleic acid within yeast during the cell growth process is similar to the *E. coli* growth process. Moreover, the trend changes in biomolecular abundance in yeast lag at about 1 h, compared to *E. coli*, which objectively indicates that *E. coli* has less time to increase generation than yeast in the synchronous culture test. All these results have implied that population heterogeneity was present during the cell growth cycle due to differences in the metabolic activities of single cells. Simultaneously, the molecular abundance assays during *E. coli* and yeast cell growth confirm the classical cell growth law, which is that cell growth determines the duration of the cell cycle, depending on the timing of the initiation and completion of log phase nucleic acid replication [19,20].

## 3. Materials and Methods

### 3.1. Bacterium Culture

In this work, we selected *E. coli* as a model for studying the abundance of biomolecules, the reason for which is that it has the advantages of rapid multiplication and easy control of culture metabolism [21,22]. Yeast was chosen as the control group. The *E. coli* and yeast were cultured into a Luria–Bertani medium for 24 h at 25 °C. Two kinds of bacteria and culture medium were obtained from the biological laboratory of Hooke Instruments Ltd. (Changchun, China) After inoculation, 3 mL of bacterial solution was taken out every 1 h, then the optical density (OD) at 600 nm was recorded using UV–Vis spectra, to estimate the growth status of the bacteria

### 3.2. Raman Detection of Bacteria during the Growth Process

Firstly, the *E. coli* and yeast were cultured at different times (0, 1, 2, 3, 4, 5, 6, 7, 8, 16, and 24 h). Subsequently, 1 µL of bacteria solution under different culture times was dropped on the aluminized glass substrate for detection using the confocal Raman spectrometer (Hooke P300, Hooke Instruments Ltd., Changchun, China), the optical system which is shown in Appendix A. To calibrate the shift of Hooke P300, we superimposed narrow-band mercury and neon lamps (Teledyne Princeton Instruments, Princeton, NJ, USA) with different wavelengths, and calibrated the relative intensity using a standard reference material (SRM 2242a). The spectral resolution of 2 cm^−1^ was measured by averaging the FWHM of the spectral lines of neon and mercury lamps. The excitation wavelength was 532 nm under 5 mW laser power. Laser excitation and Raman scattering light collection were conducted through a dry objective (100 × 0.8 NA, LMPlanFL N, Olympus, Tokyo, Japan), obtaining a spot of around 0.81 µm, smaller than the single cell size. Outstretched scan spectra were set from 400 cm^−1^ to 2000 cm^−1^, with an integration time of 3 s, and one accumulation. To reduce the effect of intracellular spatial heterogeneity on the detection of biomolecular abundance, the Raman signal was measured at three equally spaced points along the length of the *E. coli* and their average spectra were taken to represent the whole-cell Raman fingerprint spectra of *E. coli* [23].

### 3.3. Data Pre-Processing

Data pre-processing is necessary to improve the accuracy of spectral characteristic identification [24]. The Raman analysis software (Hooke intP300 V1.15) was used to pre-process the Raman data. Firstly, the Savitzky–Golay algorithm was applied to smooth and filter the data, with a window width value of 7 and a fitting order value of 3. Subsequently, the adaptive iterative reweighting penalized least squares algorithm was employed to remove the background signal of the data with a lambda value of 100, and the value of the maximum number of iterations was 15. Finally, the data were normalized using the Min–Max normalization process.

### 3.4. Target Extraction

XGBoost is a gradient-boosting algorithm with high classification, high reliability, and feature recognition ability under the integrated learning model and the tree model [25]. Therefore, it can be used to accurately identify the intracellular fingerprint information embedded in Raman spectra for biomolecular abundance detection of cell growth. The Raman characteristic peak extraction method based on XGBoost employs the average gain as the basis and determines a new characteristic splitting node by continuously iteratively learning the residuals between the predicted and true values of the cell cycle. It gives new gain information about the loss function of cell growth detection. The larger gain means that the loss is decreased, which is more useful for cell growth identification. The ℓ1 + ℓ2–norm in the loss function improves characteristic peak extraction accuracy and prevents model overfitting [26]. The XGBoost detection model is adjusted in parameter and accuracy using a grid search and cross-validation [27]. In the process of training the cell growth detection model, the characteristic gain ranking of Raman spectra is used as the output.

### 3.5. Abundance Calculation

The abundance calculation was estimated using peak area instead of peak intensity, which can improve the accuracy of abundance detection [28]. The peak area of Raman spectra can be calculated using the TPA (total peak area) method. To reduce the influence of noise fluctuations on the background from the left and right boundaries (*L*, *R*), n peak loci to both the left and the right of the characteristic peaks, for a total of 2*n* + 1 peak loci, were selected, to calculate the average value as the background. To increase the stability of the calculation method, the line between *L*, *R*, and the spectral background forms a trapezoidal region, whose area indicates the background area. The integral value between *L* and *R* represents the total area, and the formula for calculating the net peak area *A* is shown in Equation (1).
(1)$$A=\sum_{i=L}^{R}x_{i}-\left(x_{L}+x_{R}\right)\frac{\left(R-L+1\right)}{2}$$

Gaussian multi-peak fitting can realize more accurate peak area calculations when intracellular substance mixing causes mixed peak phenomena, such as superimposed distortion of spectral peaks, micro-shift of peak position, and flooding of small peaks [29]. The Gaussian fitting function was used to fit characteristic peak loci. The Gaussian multi-peak fitting problem was converted into a minimum value solution problem through a multivariate nonlinear function. Meanwhile, the nonlinear least squares as the nonlinear solution method determines the search direction using the minimum residual sum of squares. The mixed peak curve function f(x) is superimposed with several Gaussian fitting curves $g_{\mu ,\sigma }\left(x\right)$. The peak position of the Raman spectra was obtained from the multiscale wavelet transform to find the best-fit curve in the least-squares sense, as shown in Equation (2).
(2)$$f\left(x\right)=\sum g_{\mu ,\sigma }(x)=\sum_{i=1}^{N}\frac{A_{i}}{\sigma_{i}\sqrt{\pi /2}}{exp}^{-2{\left(x-\mu_{i}\right)}^{2}/\sigma_{i}^{2}}$$
where *i* is the index value of the number of Gaussian fit curves (i=1,2,…,N), *μ* is the peak position, *A* is the single-peak area, and *σ* is peak width of the characteristic peak. (We have open-sourced the Gaussian multi-peak fitting algorithm with batch support on GitHub, the source link https://github.com/lonseajim/OpenRaman, accessed on 16 January 2023).

## 4. Conclusions

In summary, we have developed a non-invasive method for detecting the abundance of biomolecules via Raman spectra combined with target extraction and multimodel fitting. The average accuracy of the XGBoost detection model was 94.4% and 98.2% for different culture times and cell cycles, respectively. The extracted characteristic peaks (760, 980, 1002 cm^−1^, and 726, 786, 1096 cm^−1^, etc.), based on the high gain information, contributed to analyzing the abundance of proteins and nucleic acids within cells. The TPA and Gaussian multi-peak fitting algorithms were developed to calculate the characteristic peak areas for the single and mixed peaks, and the fitting accuracies were better than 0.9. The abundance of protein and nucleic acid within bacteria (*E. coli* and yeast) was changed during the lag, log, and stationary phases. Notably, the detection results were consistent with the classical cell growth law, as expected. The biomolecular abundance detection method can more accurately and scientifically reveal the law of cell growth and metabolic changes at the single-cell level. This method has a guiding significance for the precise regulation of the fermentation process, to realize lower fermentation costs and higher target product yields.

## Figures and Tables

**Figure 1 molecules-29-00157-f001:**
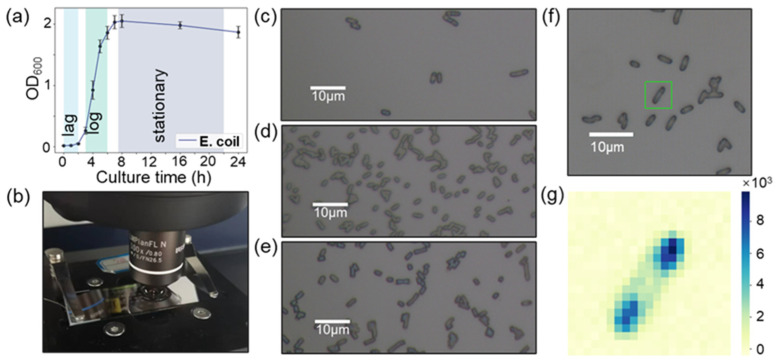
(**a**) The OD values at 600 nm as detected using UV–Vis spectra during *E. coli* growth process at different times. (**b**) Photograph of in situ Raman detection of bacterial samples using a confocal Raman spectrometer. (**c**–**e**) Bright-field images of *E. coli* during the lag phase (1 h), log phase (5 h), and stationary phase (16 h), respectively. (**f**) *E. coli* covered with a green box used for mapping detection. (**g**) Raman mapping image of single *E. coli* at 1576 cm^−1^.

**Figure 2 molecules-29-00157-f002:**
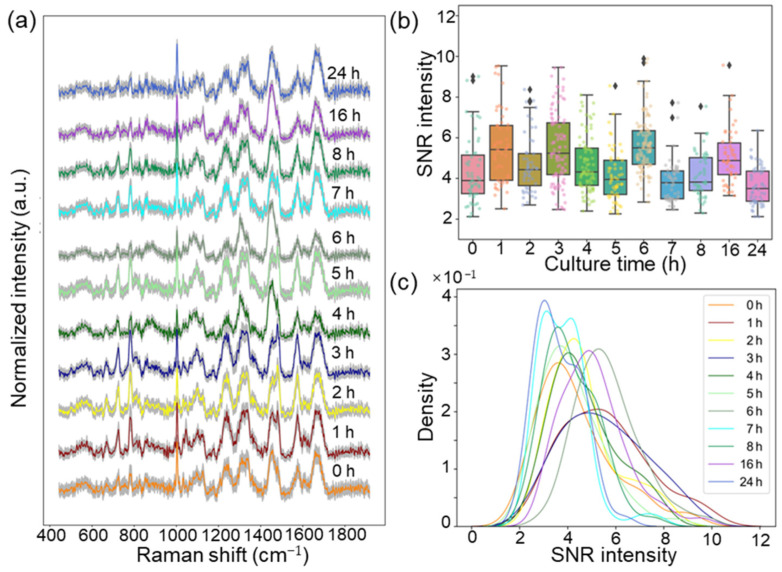
(**a**) The mean and variance Raman spectra of *E. coli* after different culture times. (**b**) The box diagram of SNR intensity under different culture times, with black diamonds indicating anomalous SNR values. (**c**) The KS normality test of SNR density curves.

**Figure 3 molecules-29-00157-f003:**
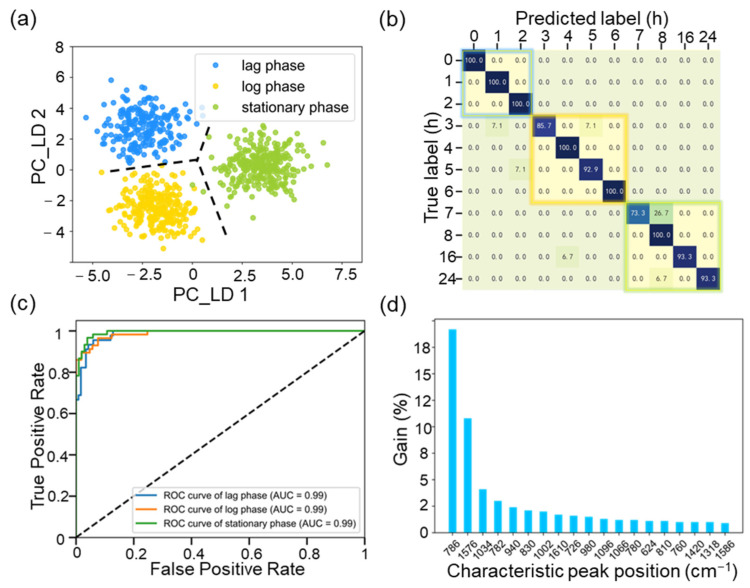
(**a**) The clustering analysis of the Raman data under lag, log, and stationary phases, respectively. (**b**) The average recognition accuracy of the cell growth detection model based on the XGBoost algorithm. (**c**) ROC curves for evaluating the sensitivity and specificity of the XGBoost model during three growth cycles. (**d**) The characteristic peak positions of the top 20 most dominant extracted through gain ranking.

**Figure 4 molecules-29-00157-f004:**
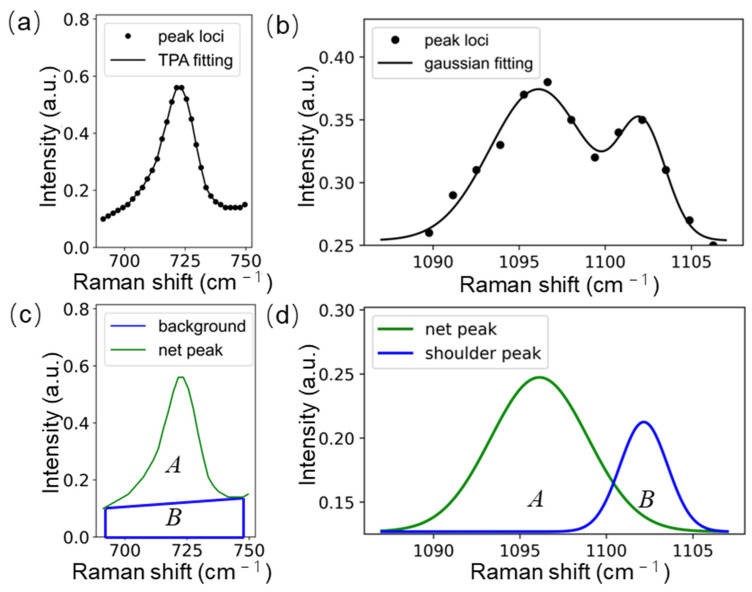
(**a**) The peak loci and the fitted curve at 726 cm^−1^ for 5 h. (**b**) The peak loci and fitted curve at 1096 cm^−1^ for 5 h. (**c**) The results of the fitting to (**a**) using TPA. (**d**) The fit to (**b**) using Gaussian multi-peak fitting.

**Figure 5 molecules-29-00157-f005:**
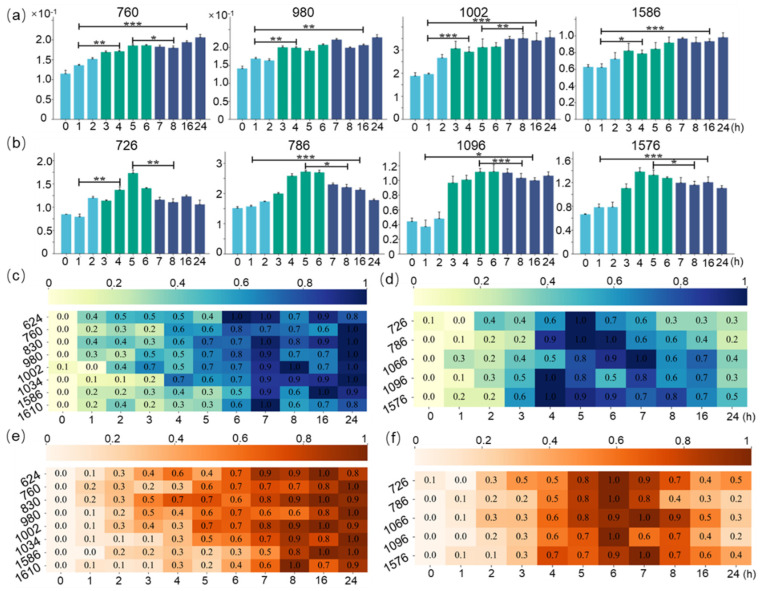
(**a**) Normal Raman intensity of Protein peaks with the lag, log, and stationary phases of cell growth process. (**b**) Normal Raman intensity of nucleic acid peaks increased with the lag, log, and stationary phases of cell growth process. (**c**,**d**) Heat map of changes in protein and nucleic acid biomolecule abundance during *E. coli* growth process. (**e**,**f**) Heat map of changes in protein and nucleic acid biomolecule abundance during yeast growth process. * represents *p* < 0.05, ** *p* < 0.01, and *** *p* < 0.001.

## Data Availability

We have open-sourced the Gaussian multi-peak fitting algorithm with batch support on GitHub, the source link https://github.com/lonseajim/OpenRaman, accessed on 16 January 2023.

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
