# Peer review of "Non-Invasive Detection of Biomolecular Abundance from Fermentative Microorganisms via Raman Spectra Combined with Target Extraction and Multimodel Fitting"

_molecules, 2023, doi:10.3390/molecules29010157_

Round 1

Reviewer 1 Report

Comments and Suggestions for Authors

Reviewer comments

(1)      In the third paragraph of the Introduction, what does "peak position micro-shift" mean?

(2)      In section 2.2, "The excitation wavelength was set as 532 nm under 5 mW laser power. Laser excitation and Raman scattering light collection were conducted through a dry objective", what spectral resolution is used? What are the shift and intensity calibration methods for Raman spectrometers?

(3)      In section 2.2, "their average spectra were taken to represent the whole-cell Raman fingerprint spectra of E. coli", why only use the Raman spectra of the fingerprint area?

(4)      In section 2.5, what is the TPA method? Please clarify it.

(5)      In section 3.3, "the goodness-of-fit (R2) is 0.94 ± 0.05, and B is the shoulder peak area at 1102.2 cm-1 in the mixed peak.", what is the spectral resolution, and does it make sense to keep the wavenumber one decimal place?

(6)      Please add the highlights of the manuscript.

Comments on the Quality of English Language

Well

Reviewer 2 Report

Comments and Suggestions for Authors

The authors have presented a non-invasive method for detecting the abundance of biomolecules of fermentation microorganisms via Raman spectra combined with target extraction and multi-peak fitting.  Data has demonstrated the population heterogeneity including protein and nucleic acid abundance during lag, log, and stationary phases of cell growth in E. coli and yeast. There are a few minor issues that authors should take into account before this manuscript is published:

What is the spatial resolution of Raman imaging in Figure 1?  Is it enough to detect the distribution of E. coli and yeast cells and to guide for Raman spectra measurement “at three equally spaced points along the length of a bacteria”. Could authors provide some relevant values? 

2. In line 197, “In summary, the SNR result shows a more stable distribution, ensuring that the expected detection results are not affected by SNR.” Could authors clarify "more stable distribution" compared to what? 

Comments on the Quality of English Language

“Line 44: Raman spectroscopy is a fingerprint spectrum, that provides relatively high sensitivity and is widely applied in the biological field [6-7].” Raman spectroscopy is a technique, not a spectrum. This sentence needs to be corrected.  

Several sentences require grammar check or rephrasing for clarity, for example, Line 188: The Raman detection was performed using the Raman spectrometer self-built, whose optical system information is displayed in Figure A2,” and “Line 193: the 193 spectral SNR is concentrated between 3 and 7.” Authors should check the language more thoroughly. 
